# Takotsubo Cardiomyopathy in Cancer Patients Treated with Immune Checkpoint Inhibitors: A Systematic Review and Meta-Summary of Included Cases

**DOI:** 10.3390/cancers15092637

**Published:** 2023-05-06

**Authors:** Ioannis P. Trontzas, Ioannis A. Vathiotis, Konstantinos G. Kyriakoulis, Amalia Sofianidi, Zoi Spyropoulou, Andriani Charpidou, Elias A. Kotteas, Konstantinos N. Syrigos

**Affiliations:** 1Department of Pathology, Yale University School of Medicine, New Haven, CT 06519, USA; 2Oncology Unit, Third Department of Internal Medicine, Sotiria General Hospital for Chest Diseases, National and Kapodistrian University of Athens, 11527 Athens, Greece

**Keywords:** takotsubo cardiomyopathy, takotsubo syndrome, broken heart syndrome, TTS, immunotherapy, immune checkpoint inhibitors, immune-related adverse events, cancer

## Abstract

**Simple Summary:**

Takotsubo syndrome (TTS), also known as “broken heart” syndrome, is a rare but serious heart event associated with physical or emotional stress. Patients usually complain about intense chest pain and difficulty in breathing, resembling acute myocardial infarction. Cancer patients are more likely to suffer from TTS and the condition has been linked with the use of several anticancer remedies, such as chemotherapy and targeted therapy. Immunotherapy is a new, viable option for many patients and is currently used in the management of several cancers. There are emerging reports of TTS in cancer patients treated with immunotherapy; however, its causality on TTS development remains uncertain. As TTS may be life-threatening and impact anticancer management, it is crucial to identify any possible association with immunotherapy. In this literature review, we tried to explore the extent of the condition in immunotherapy-treated patients, and to provide information regarding accurate diagnosis, management, and outcomes.

**Abstract:**

Background: There are emerging reports of Takotsubo syndrome (TTS) in cancer patients treated with immune checkpoint inhibitors (ICIs); however, the association of the two remains uncertain. Methods: A systematic literature review was performed in the PubMed database and web sources (Google Scholar) according to the Preferred Reporting Items for Systematic reviews and Meta-analyses (PRISMA) guidelines. Case reports/series or studies including cancer patients treated with ICIs and presenting with TTS were considered. Results: Seventeen cases were included in the systematic review. Most patients were males (59%) with median age of 70 years (30–83). Most common tumor types were lung cancer (35%) and melanoma (29%). Most patients were on first-line immunotherapy (35%) and after the first cycle (54%) of treatment. The median time on immunotherapy at the time of TTS presentation was 77 days (1–450). The most used agents were pembrolizumab and the combination of nivolumab–ipilimumab (35%, respectively). Potential stressors were recognized in 12 cases (80%). Six patients (35%) presented with concurrent cardiac complications. Corticosteroids were used in the management of eight patients (50%). Fifteen patients (88%) recovered from TTS, two patients (12%) relapsed, and one patient died. Immunotherapy was reintroduced in five cases (50%). Conclusion: TTS may be associated with immunotherapy for cancer. Physicians should be alert for TTS diagnosis in any patient with myocardial infarction-like presentation under treatment with ICIs.

## 1. Introduction

Since the approval of the cytotoxic T-lymphocyte associated protein-4 (CTLA-4) inhibitor ipilimumab for the treatment of patients with metastatic melanoma in 2011 [1,2], immune checkpoint inhibitors (ICIs) have revolutionized the management of cancer patients. ICIs, including programmed death protein receptor-1 (PD-1), programmed death ligand-1 (PD-L1), and CTLA-4 inhibitors, are monoclonal antibodies used for the treatment of various malignancies across different clinical settings and have significantly improved the cancer-related outcomes [3]. However, immune activation may lead to a novel spectrum of toxicities, collectively known as immune-related adverse events (irAEs) [4]. These adverse events may involve any organ system and have variable clinical presentations [4].

Immune-related cardiovascular toxicity (irCVT) is rare but constitutes a serious concern as it can be life threatening [5]. In a recent systematic review and meta-analysis, the incidence of irCVT among different studies was 1.3% [6]. Myocarditis represents the most frequent irCVT, accounting for 45% of observed events in case reports/series [5]. Other irCVTs, including arrhythmias (atrial fibrillation, supraventricular tachycardia, and bradyarrhythmias), pericarditis, and pericardial effusion, are less frequent [5]. IrCVT has been reported to occur after single-agent immunotherapy (pembrolizumab, nivolumab), but the incidence is higher with the combined use of different ICIs (nivolumab and ipilimumab compared to nivolumab alone) [7]. As ICIs are considered drugs with potency to induce myocarditis, which may be life-threatening, treatment with high-dose corticosteroids is indicated upon clinical suspicion [7].

Takotsubo cardiomyopathy (TTS), also known as “broken heart” syndrome, is characterized by acute chest pain, dyspnea, or syncope, usually following emotional or physical stress that may resemble acute myocardial infarction (MI) [8]. Distinct echocardiography findings include apical ballooning and midventricular, basal, or focal wall motion abnormalities; a normal coronary angiography (CAG) usually distinguishes TTS from MI scenarios by excluding true type-1 atherosclerotic MI [8]. There is no consensus for TTS diagnostic criteria, since many research groups have proposed different diagnostic approaches based on the respective ethnic phenotypes [8]. The Mayo Clinic Diagnostic Criteria are the most widely known [9], while, in 2018 the InterTAK Diagnostic Criteria were proposed in an effort to incorporate the discrepancies and provide a worldwide consensus [8]. Management is initially directed to acute myocardial ischemia, since the syndrome mimics an acute coronary syndrome, and then supportive measures are indicated with the intention to relieve heart failure symptoms and treat any underline triggers [10]. After the acute phase, the syndrome tends to be spontaneously resolved, but long-term complications may invariably persist [10].

TTS has been associated with cancer and the use of several anticancer agents; however, its association with immunotherapy remains vastly unknown [8]. To date, TTS has only been linked with the use of ICIs in scarce case reports [11]. As a result, although the European Society of Medical Oncology (ESMO) has identified a possible correlation of TTS with immunotherapy, TTS is not currently included in the list of irCVTs [7]. Other Oncology societies (American Society of Clinical Oncology (ASCO), National Comprehensive Cancer Network (NCCN)) have also not incorporated TTS management in irCVTs management guidelines. Indeed, the association of TTS with immunotherapy is difficult not only due to the difficulties in diagnosis and the rarity of the condition but also because other cardiac events (e.g., myocarditis) tend to simultaneously present in many cases [11].

Increasing use of ICIs would be expected to raise the incidence of a potentially immune-related TTS. Since the condition may be life-threatening and it would impact the anticancer management and outcomes, investigation of its correlation with immunotherapy is crucial, with subsequent implications on accurate diagnosis and management.

As the association of TTS with immunotherapy for cancer remains uncertain, we were prompted to investigate the reported extent of the condition. We herein conducted a systematic literature review aiming to identify relevant reports of TTS in patients treated with ICIs. We then sought to summarize the clinicopathologic characteristics and outcomes of included cases. Moreover, we add to current knowledge with the inclusion of one relevant and previously unpublished case, managed in our hospital.

## 2. Materials and Methods

### 2.1. Design 

The study design was discussed and agreed upon in advance by all authors. A PICO-S (Population, Intervention, Comparison, Outcomes, Study selection) [12] approach for the facilitation of our search strategy was used. The population included cancer patients, while intervention was treatment with ICIs. A comparison group was not included. The outcome of interest was TTS, and we performed a search to identify any type of relevant study. The PICO-S approach is demonstrated in Appendix A.

### 2.2. Reporting and Protocol Registration

The systematic review was performed in line with the Preferred Reporting Items for Systematic reviews and Meta-Analyses (PRISMA) guidelines [13]. The PRISMA 2020 checklist for the present systematic review is presented in Appendix A. The protocol was registered in the PROSPERO international prospective register of systematic reviews (ID: 348753).

### 2.3. Search Strategy

A systematic search of PubMed was performed until 13 March 2023 using the following search algorithm: (“takotsubo cardiomyopathy” OR “takotsubo syndrome” OR “TTS” OR “stress-induced cardiomyopathy” OR “broken-heart syndrome” OR “non-ischemic cardiomyopathy” OR “transient cardiomyopathy”) AND (“immunotherapy” OR “cancer immunotherapy” OR “immune checkpoint inhibitors” OR “checkpoint blockade” OR “anti-PD-1” OR “PD-1” OR “programmed death ligand-1” OR “PD-L1” OR “anti-PD-L1” OR “programmed death ligand-1” OR “CTLA-4” OR “anti-CTLA-4” OR “cytotoxic T-lymphocyte antigen-4” OR “pembrolizumab” OR “nivolumab” OR “ipilimumab” OR “durvalumab” OR “avelumab” OR “cemiplimab” OR “atezolizumab”). An additional search with the same algorithm was performed in Google Scholar. Reference lists of eligible articles were also systematically searched for any additional reports (‘snowball procedure’).

### 2.4. Cases and Studies Selection

Included cases were retrieved by four reviewers (I.P.T., K.G.K, A.S., and Z.S.) who worked independently. Disagreements were resolved after consensus with two senior authors (E.A.K and K.N.S.) was reached. Eligible items were full-text articles written in the English language. Isolated case reports, case series, and retrospective studies of TTS events in patients treated with ICIs were assessed for inclusion. Non-English articles, publications not providing full manuscript, and not-peer-reviewed reports were excluded.

### 2.5. Data Extraction and Tabulation

Five reviewers (I.P.T., I.A.V., K.G.K., A.S. and Z.S.) worked independently for extraction and tabulation of data regarding the main characteristics of patients described in the included articles (age, gender, tumor type, potential stressors, type of immunotherapeutic agent, line of therapy, cycles of therapy, concurrent/previous systemic therapies, time on ICI treatment at TTS presentation, time from last ICI treatment, symptoms at TTS presentation, echocardiography findings, CAG findings, electrocardiography (ECG) findings, cardiac enzymes elevation, concurrent cardiac complications, management interventions, outcome, and reintroduction of immunotherapy). History of cardiovascular (CVS) disease, active smoking, and any physical or emotional stress were regarded as potential stressors, in line with the available literature [14].

Communication with the authors of eligible reports was sought with the intention of individual patients’ demographic/outcome data analysis. Two author teams provided raw data after correspondence [15,16]. As a result, details regarding the outcomes of TTS on overall anticancer management, on survival of affected patients, and several baseline clinical characteristics (e.g., ECOG performance status, non-CVS comorbidities) were not included in descriptive analysis. The rest of the clinicopathological characteristics were synthesized based on the available evidence of published sources.

## 3. Results

### 3.1. Literature Search and Selection of Reports

Of the seventy-seven articles initially retrieved through the literature search, fifteen met all eligibility criteria and were included in the systematic review [15,16,17,18,19,20,21,22,23,24,25,26,27,28,29]. One additional unpublished case from the Third Department of Internal Medicine, Sotiria General Hospital for Chest Diseases, National and Kapodistrian University of Athens, Athens, Greece, was also included. Two retrospective studies, reporting a further seventeen cases, were excluded due to possible overlap with other cases included in our study, after unsuccessful communication with corresponding authors [30,31]. Overall, seventeen cases were included in the systematic review and descriptive analysis. The PRISMA 2020 flow diagram for systematic reviews and meta-analyses of studies selection is illustrated in Figure 1.

### 3.2. Baseline Clinicopathologic Characteristics

Baseline characteristics of all included cases are presented in Table 1. Descriptive analysis of each characteristic was based on the total number of cases (*N*) providing the respective data. Out of 17 patients, 10 (59%) were males with median age of 70 years (30–83). Tumor type included lung cancer (35% (*n* = 6, *N* = 17)), melanoma (29% (*n* = 5)), hepatocellular carcinoma (12% (*n* = 2)), renal cell carcinoma (12% (*n* = 2)), head and neck squamous cell carcinoma (6% (*n* = 1)), and breast cancer (6% (*n* = 1)). Evidence regarding line of treatment and cycle of therapy at TTS presentation was provided in 13 cases. Among them, 10 (77%) were on first-line treatment, 2 (15%) were on second-line treatment, and 1 (8%) was on adjuvant treatment. Seven patients (54%) presented with TTS after the first immunotherapy cycle, one (8%) after the third cycle, two (15%) after the fourth cycle, and three (23%) in later cycles (seventh and beyond). The median time on ICI treatment at the time of TTS was 77 days (1–450 *(N* = 15)), and median time from last ICI administration was seven days (1–136 (*N* = 10)). Ten patients (59% (*N* = 17)) were treated with monotherapy, while the rest received combination immunotherapy. With respect to ICI type, seven patients (41% (*N* = 17)) were treated with anti-PD-1 agents (pembrolizumab, 35% (*n* = 6); nivolumab, 6% (*n* = 1)), six (35%) with anti-PD-1/anti-CTLA-4 combination (nivolumab/ipilimumab), two (12%) with anti-PD-L1 monotherapy (atezolizumab), and two with anti-CTLA-4 monotherapy (ipilimumab) or anti-PD-L1/anti-CTLA-4 combination (durvalumab/tremelimumab) (6%, respectively). Five patients (31% (*N* = 16)) were receiving concurrent systematic treatment and six (38%) had received other systemic treatment prior to ICIs, including chemotherapy (cisplatin, carboplatin, pemetrexed, gemcitabine, paclitaxel, anthracycline-based regimens, 5-FU, and vinorelbine) and targeted therapy (trastuzumab, bevacizumab, cetuximab, axitinib, and sorafenib). Potential stressors were identified in 12 patients (80% (*N* = 15)), and those included history of active smoking, CVS disease, and physical (concurrent cardiac complications, other irAEs, infection, and infusion reactions) or emotional stress (intense stress on the ground of major depression).

### 3.3. Clinical Characteristics and Outcomes of Takotsubo Syndrome

Clinical characteristics at TTS presentation, management, and outcomes are presented in Table 2. Main symptoms at presentation included dyspnea (67% (*n* = 10, *N* = 15)) with or without respiratory distress, chest pain (47% (*n* = 7)), and weakness/fatigue (13% (*n* = 2)). Other symptoms were nausea/vomiting, gastrointestinal symptoms (diarrhea, abdominal cramping), confusion, diaphoresis, palpitations, wheezing, fever, and generalized pain. One patient presented with signs of cardiogenic shock. All included reports provided echocardiographic findings at admission, with the most prominent features being apical ballooning (76% (*n* = 13)) and hyperkinesis of not affected cardiac segments (29% (*n* = 5)), mainly of basal and septal walls. Notably, two patients (12%) presented with atypical TTS with akinesia of middle and basal segments. Moreover, 12 patients (92% (*n* = 13)) had diminished left ventricular ejection fraction (LVEF). All patients with available data demonstrated ECG changes suggestive of MI (ST elevation, T-wave inversion (*N* = 16)), had abnormalities of cardiac enzymes (elevation of troponin and/or brain natriuretic peptide (*N* = 16)), and CAG without evidence of acute coronary obstruction (*N* = 15). In addition, six patients (35% (*N* = 17)) demonstrated concurrent cardiac complications (myocarditis, pericarditis/pericardial effusion, myopericardial carcinomatous infiltration, malignant embolization of coronary arteries, and ventricular tachycardia) and ten (62% (*N* = 16)) had concurrent irAEs (pneumonitis, diabetic ketoacidosis, colitis, hepatitis, nausea/vomiting, skin toxicity, nephritis, and infusion reaction). Data regarding TTS management were provided in 16 cases. Noteworthy, only eight patients (50%) were managed with high-dose corticosteroids. Other remedies included MI and/or heart failure management, including use of β-blockers, angiotensin-converting enzyme inhibitors, and diuretics, other supportive measures, such as oxygen therapy, IV fluids, and administration of anxiolytics, and treatment of the underlying trigger (e.g., insulin for diabetic ketoacidosis). Regarding outcomes, TTS resolved completely in 15 cases (88% (*N* = 17)). In three patients, resolution of apical ballooning followed several weeks after clinical improvement or did not resolve at all (sustained apical ballooning on echocardiography one year later), and one patient died due to ir-CVT (including TTS) and cancer complications. The median time to resolution was 8 days (3–35 (*N* = 9)). ICIs were reintroduced in five patients (50% (*N* = 10)) after the index event. It should be noted that two patients experienced TTS relapse: the first event occurred right after the reintroduction of the ICI (four days), while the second occurred seven months after the last ICI dose.

## 4. Discussion

TTS as a potential complication of anticancer treatment is an uncommon but potentially life-threatening condition. The widespread use of ICIs has brought up a wide spectrum of irAEs. However, a direct link between ICIs’ use and TTS remains uncertain.

Indeed, an association between TTS and ICIs is difficult to establish as the diagnosis of the former is challenging and may overlap with other forms of cardiotoxicity (e.g., myocarditis, pericarditis) [17,22,27]. Moreover, history of malignancy appears to be an independent risk factor for TTS, possibly due to intense physical and emotional stress in patients with cancer [32]. Thus, diagnosis of an immune-related TTS is complex, and case-by-case examination is necessary. Notably, cancer patients experience worse TTS outcomes compared with those without malignancy [33], which highlights the need for thorough reporting and careful monitoring of cancer-associated TTS.

To the best of our knowledge, this is the first systematic review specifically designed to retrieve published cases of cancer patients treated with ICIs that developed TTS. Other reviews on the topic have investigated either the impact of anticancer agents to the development of TTS [11,32,34] or the overall cardiotoxic effect of ICIs [5,35,36,37,38,39]. Our systematic search and study design allowed us to retrieve 17 cases, the most reports recruited up to date in the literature, and to meticulously describe several TTS characteristics. Despite that, the number of recruited cases remains small, which was an anticipated limitation of our study. Another limitation lies in the poor evidence for a direct association of TTS with ICIs in the included cases. We investigated several validated algorithms (e.g., Naranjo, Yale, Jone’s, Karch, Begaud, ADRAC, WHO-UMC, Bradford-Hill) [40] for the assessment of probability of adverse drug reaction; however, their application was not feasible either due to lack of sufficient evidence, lack of control group, or due to structural issues of the algorithms. As a result, we acknowledge that the association of ICIs and TTS in the included cases does not necessarily imply causation.

Noteworthy, one of the patients included in our study presented with TTS soon after the first introduction of ICIs (nivolumab–ipilimumab for melanoma) and relapsed soon after the second infusion, which led to immunotherapy discontinuation. This case may imply direct causality of ICIs to the TTS event. Another patient treated with pembrolizumab for lung cancer relapsed with TTS; however, in that case, the relapse occurred seven months after last ICI infusion; thus, other underlying triggers may be implicated as well.

We believe that the challenges in TTS diagnosis may mask the true impact of the condition in immunotherapy-treated patients. In respect to that, seventeen more cases have been described in two retrospective studies [30,31], which were excluded from our systematic review as described earlier. In the first study, Ederhy et al. [30] performed an analysis using data from the World Health Organization (WHO) global database of individual case safety reports. The authors compared the proportion of TTS in patients receiving ICI vs. those receiving paracetamol, adrenaline, and venlafaxine (controls). ICIs were associated with a higher proportion of TTS; this analysis using control groups may enhance the hypothesis of causal relation between ICIs and TTS. In the second study, Escudier et al. [31] demonstrated that TTS may not be extremely rare among patients suffering from ir-CVT after examining the cardiotoxicity cases identified in the databases of two cardio-oncology clinics and in the published literature until 2017 (TTS was present in four out of 29 ir-CVT cases). This emerging evidence alerts us to the possible association of the condition with ICIs’ use, so we encourage our peers in the clinical setting to rigorously report similar cases.

Furthermore, we tried to investigate possible stressors triggering TTS in those patients. We observed that other cardiac complications coexisted in several cases [15,17,20,22,27], mainly myocarditis. As has been previously discussed, ICIs are recognized as therapeutic agents with strong potential to induce myocarditis [5,6]. Since presentation of myocarditis may resemble, or overlap with, TTS [5,17,38,39], a cardiac magnetic resonance imaging (MRI) with gadolinium enhancement or even a myocardial biopsy are needed for the exclusion of immune-related myocarditis. Differentiation of the two clinical entities is crucial, as the presence of myocarditis alerts us to urgent ICIs’ discontinuation and corticosteroids administration [7]. In our review, we identified five cases (29%) with concurrent myocarditis (possible or diagnosed) [13,15,20,22,25] and one with myopericardial effusion due to malignant infiltration [18]. However, out of the remaining eleven cases not reporting concurrent cardiac complications, the necessary diagnostic workup for myocarditis exclusion was performed only in two patients. One more case reported exclusion of myocarditis based on MRI-findings; however, relevant data to interpret the results were not provided. Based on the cases included in our review, there is insufficient evidence to support whether TTS is a phenotype of immunotherapy-related myocarditis or if it is a distinct cardiotoxic effect. A larger sample size following the necessary diagnostic approaches for the precise exclusion of other concurrent irCVTs (mainly myocarditis) would provide better insight.

Other possible stressors included the presence of non-cardiac irAEs [15,17,18,19,23,26,28,29], infections [21], and infusion reactions [16]. Other risk factors, such as active smoking and cardiovascular comorbidities, have been previously described [14] and were also identified in our study [17,18,19]. In addition, several anticancer drugs (e.g., trastuzumab, anthracyclines, axitinib, 5-FU, etc.), of which the association with TTS has been previously described [11,32], may also be also implicated in triggering TTS in some of the cases [20,25,26,29]. The abundance of potential stressors suggests that TTS may not be a direct outcome of immune-related toxicity, and that therapy-induced immune stimulation may aggravate other underlying mechanisms.

Regarding time of TTS presentation, there are lines of evidence in our review (median time on immunotherapy at the time of TTS presentation was found to be 77 days) and in other studies [11,29] denoting that immune-related TTS represents a rather delayed form of cardiotoxicity. However, we believe that these observations may be biased from the small sample size. This is highlighted in cases where TTS occurred after hours or several days [15,16,17,18,24,27,29] from the last ICI administration.

In addition, a detailed description of other critical parameters, such as the impact of corticosteroids in the management of TTS or that of TTS in anticancer outcomes, was not provided, as respective data were not mentioned in most cases and correspondence with authors was unsuccessful.

Better understanding of the underlying mechanisms would provide an insight into TTS and its potential association with ICIs. Although pathophysiology of TTS remains vastly unknown, various mechanisms have been proposed in the literature, including catecholamine excess, coronary vasospasm, microvascular dysfunction, and upregulation of certain cardiac genes [32,41]. The hormonal influence on TTS is supported by the higher frequency of the condition in women (90% of the cases) and in older patients (80% of the cases are older than 50 years old) [8]. Interestingly, a lower incidence of TTS in females with cancer has been reported in several studies [11,42], possibly attributed to the different underlying mechanisms of the condition compared to TTS in non-cancer patients. Lower incidence of TTS on females is consistent with our findings (59% male gender). The different underlying mechanisms of TTS in cancer patients may be at least partially attributed to anticancer drugs effects. Several mechanisms of immunotherapy-related TTS have been proposed in the literature. It has been suggested that autoreactive T-cells may cross-react with myocardium [35], that ICIs may directly act on coronary arteries, leading to coronary vasospam in multiple areas, or that an increased release of catecholamines is triggered from the adrenal glands and postganglionic sympathetic nerves upon immune blockade, thus inducing myocardial response [43,44]. A more thorough assessment of the reported cases is needed for more solid conclusions. A comprehensive illustration summarizing the proposed series of events leading to the development of TTS in ICI-treated cancer patients is provided in Figure 2.

With the expanding use of ICIs, physicians are more likely to encounter TTS cases with an immune component in the future. Some critical questions include the following: (i) Is there a role of immunosuppressant therapy in the management of an immune-related TTS? (ii) Is it safe to reintroduce ICIs after a life-threatening but reversible event? (iii) What is the impact of TTS in the cancer-related outcomes. As a result, management of such cases should be individualized upon physicians’ and tumor boards’ discretion.

## 5. Conclusions

Malignancy represents an independent risk factor for TTS. Although accumulating reports have linked TTS with ICIs treatment, a causal association between the two has not yet been established. Physicians should be alert for TTS diagnosis in any patient with MI-like presentation under treatment with ICIs. Moreover, there is an unmet need regarding the management of immune-related TTS. As such, therapeutic decisions should currently be individualized; thorough reporting of similar cases will provide an insight into the underlying mechanisms with implications for accurate diagnosis and effective treatment.

## Figures and Tables

**Figure 1 cancers-15-02637-f001:**
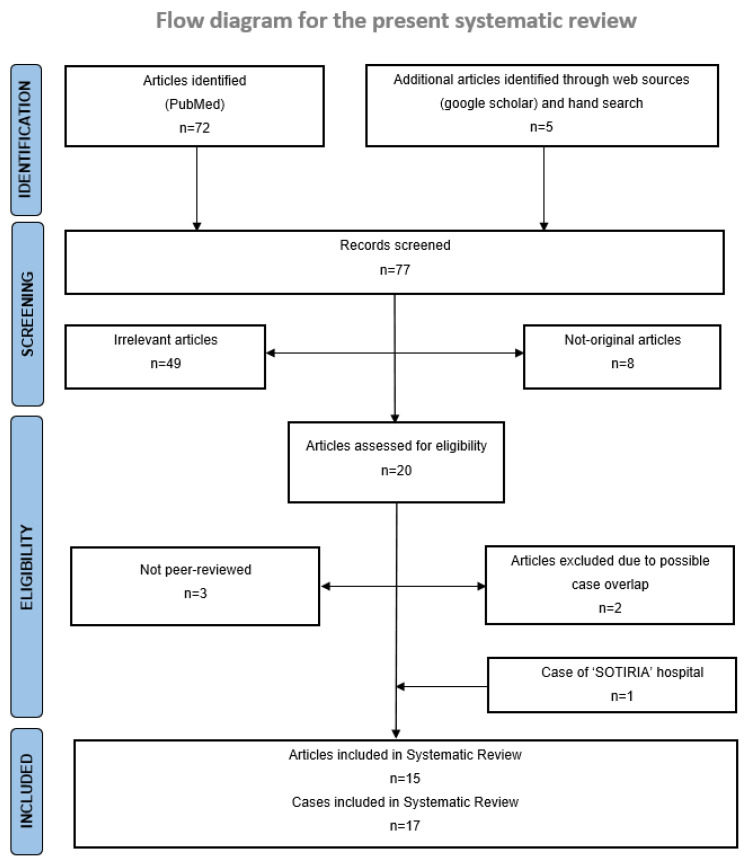
PRISMA 2020 flow diagram for systematic reviews and meta-analyses of studies selection [13]. For more information, visit: http://www.prisma-statement.org/.

**Figure 2 cancers-15-02637-f002:**
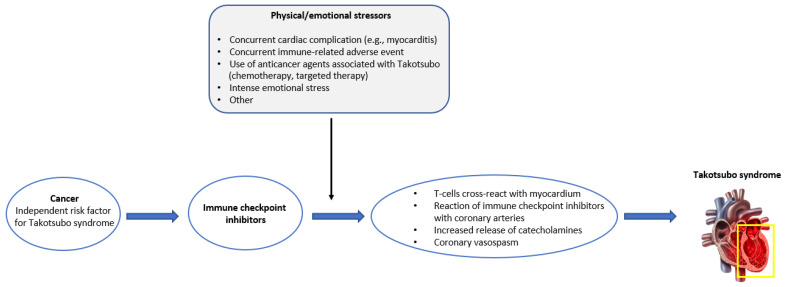
Graphical illustration of the proposed series of events leading to Takotsubo syndrome in cancer patients treated with immune checkpoint inhibitors.

**Table 1 cancers-15-02637-t001:** Baseline characteristics and immunotherapy setting.

Ref	Gender, Age	Tumor Type (Stage)	Potential Stressors *	Line/Cycle of Therapy	ICIs Targets (Agents)	Concurrent/Previous Systemic Therapies	Time on Treatment at TTS Presentation (Days)	Time from Last ICI Treatment (Days)
Tan et al., 2020 [15]	M, 62	HCC (IV)	Active smoking	1st line, C1	PD-1 (nivolumab)	-	21	21
Oldfield et al., 2020 [16]	M, 76	Melanoma (IV)	Underlying DKA, CVS comorbidities (hypertension, diabetes mellitus, hyperlipidemia)	1st line, 1st event after C1, 2nd event after C2	PD-1 (nivolumab), CTLA-4 (ipilimumab)	-	NR	2nd event 4 days after C2 (1st event NR)
Geisler et al., 2015 [17]	F, 83	Melanoma (IV)	Hypertension	1st line, C4	CTLA-4 (ipilimumab)	-	≈84	≈21
Elikowski et al., 2018 [18]	M, 30	NSCLC (IV)	Cardiac carcinomatous infiltration, carcinomatous embolization of coronary arteries	Patient received ICIs in 1st line, TTS presented after 2nd line ChT	PD-L1 (durvalumab), CTLA-4 (tremelimumab)	Cisplatin/gemcitabine (1st line in combination with ICIs), carboplatin/paclitaxel (2nd line)	NR	NR
Khan et al., 2020 [19]	F, 57	NSCLC (IV)	Underlying pneumonia	1st line, C4	PD-1 (pembrolizumab)	Carboplatin/pemetrexed (combination with ICIs)	≈77	14
Tsuruda et al., 2021 [20]	M, 75	NSCLC	Myocarditis	1st line, C1	PD-1 (pembrolizumab)	Adjuvant cisplatin/vinorelbine 6 months ago	136	136
Serzan et al., 2021 [21]	F, 66	Melanoma (I)	-	Adjuvant, C7	PD-1 (nivolumab), CTLA4 (ipilimumab)	-	≈112	NR
Ederhy et al., 2017 [22]	M, 45	Melanoma (advanced)	NR	Line NR, C1	PD-1 (nivolumab), CTLA4 (ipilimumab)	-	5	5
	M, 77	Melanoma (advanced)	NR	Line NR, C3	PD-1 (nivolumab), CTLA-4 (ipilimumab)	-	65	NR
Okamatsu et al., 2020 [14]	F, 76	NSCLC (IIIC)	Infusion reaction	1st line, C1	PD-1 (pembrolizumab)	-	1 (6 h after C1)	1 (6 h after C1)
Anderson & Brooks, 2016 [23]	F, 56	HER2 breast cancer (IV)	Colitis	1st line	PD-1 (pembrolizumab)	Trastuzumab along with ICI, previous adjuvant treatment with anthracycline based ChT and trastuzumab (about 8 months before ICI)	≈247	NR
Schwab et al., 2019 [24]	M, 69	HNSCC (IV)	-	2nd line, C7	PD-1 (nivolumab), CTLA-4 (ipilimumab)	Previous ChT with cisplatin, 5-FU, cetuximab (at least 1 month before ICIs)	≈450	NR
Camastra et al., 2020 [13]	M, 70	Lung cancer	Possible myocarditis, immune-induced nausea/vomiting	Line NR, C1	PD-L1 (atezolizumab)	Previous ChT	7	7
Norikane et al., 2020 [25]	M, 73	RCC (advanced)	Myocarditis	NR	PD-1 (nivolumab), CTLA-4 (ipilimumab)	NR	7	7
Singhal et al., 2022 [26]	F, ≈80	HCC (IV)	Underlying DKA, hypertension	2nd line	PD-L1 (atezolizumab)	Bevacizumab along with ICI, previous line with multi-TKI (sorafenib) at least 6 months before	≈180	NR
Airo et al., 2022 [27]	M, 49	RCC (IV)	-	1st line, C1	PD-1 (pembrolizumab)	VEGFR-TKI (axitinib) along with ICI	6	6
‘Sotiria’ case	F, 74	NSCLC (IIIC)	Active smoking, major depression, CVS comorbidities (PAD, hypertension, CAD)	1st line, C8	PD-1 (pembrolizumab)	-	≈175 (2nd event 360 days after C1)	7 (2nd event 180 days after C8)

C: Cycle; CAD: Coronary artery disease; ChT: Chemotherapy; CTLA-4: Cytotoxic T-lymphocyte associated protein-4; CVS: Cardiovascular; DKA: Diabetic ketoacidosis; F; Female; HCC: Hepatocellular carcinoma; HNSCC: Head and neck squamous cell carcinoma; ICIs: Immune checkpoint inhibitors; M: Male; NR: Not reported; NSCLC: Non-small cell lung cancer; PAD: Peripheral arterial disease; PD: Programmed death receptor; PD-L1: Programmed death ligand-1; RCC: Renal cell carcinoma; TKI: Tyrosine kinase inhibitor; TTS: Takotsubo syndrome; VEGFR: Vascular endothelia growth factor receptor; * Potential stressors include known cardiovascular comorbidity, active smoking, emotional, or physical stress. -: No.

**Table 2 cancers-15-02637-t002:** Characteristics and outcomes of Takotsubo syndrome.

Ref	Clinical Presentation	Diagnostic Workup					Concurrent Cardiac Complications	Other irAEs	Management	Outcome	
		Echocardiography	↓LVEF	ECG Findings	↑Cardiac Enzymes	CAG				TTS Outcome	Re-Introduction of ICIs
Tan et al., 2020 [15]	Chest pain, nausea, vomiting	Apical ballooning	+	ST elevation (V5-6, II-III), RBBB, LAFB	+ (Trop & BNP)	Non-obstructive CAD	Myopericarditis, VT	Pneumonitis	Corticosteroids	Resolution (clinical on 3 days, imaging on 42 days)	-
Oldfield et al., 2020 [16]	Chest pain, diaphoresis	Apical ballooning, hyperkinetic basal and mid segments	+	ST elevation (V2-6)	+ (Trop)	Non-obstructive CAD	− (no biopsy or MRI performed for exclusion of myocarditis)	DKA, colitis, hepatitis	MI management, corticosteroids	Resolution (but 2nd event after C2)	− (interrupted after 2nd event)
Geisler et al., 2015 [17]	Chest pain, dyspnea	Apical ballooning, hyperkinetic base, and septum	+	ST elevation (I, V2-3), VT	+ (Trop)	Non-obstructive CAD	− (no biopsy or MRI performed for exclusion of myocarditis)	Colitis, pruritus, malaise	β-blocker	Clinical resolution (5 days, but apical ballooning persisted)	NR
Elikowski et al., 2018 [18]	Dyspnea, weakness	LV contractility disturbances typical of apical TTS	NR	Negative T-waves (V1-6)	+ (Trop & BNP)	NR	Myopericardial malignant infiltration, malignant embolization of coronary arteries	NR	HF management	Resolution, but patient died after few days	-
Khan et al., 2020 [19]	Chest pain, palpitations, dyspnea	Hypokinesia of septum and anterior wall with sparing of apical and basal segments (atypical TTS)	+	Sinus tachycardia	+ (Trop)	Chronic RCA obstruction	− (no biopsy or MRI performed for exclusion of myocarditis)	-	HF management	Resolution	NR
Tsuruda et al., 2021 [20]	NR	Apical ballooning	NR	T-wave inversion, QT prolongation	+ (Trop)	Not done	Myocarditis, pericardial effusion (cardiac MRI with late gadolinium enhancement and suggestive endomyocardial biopsies)	-	Corticosteroids	Death	-
Serzan et al., 2021 [21]	Dyspnea, generalized pain	Apical ballooning, hyperdynamic basal LV segments	NR	Sinus tachycardia, inferolateral T-wave inversions	+ (Trop & BNP)	Non-obstructive CAD	− (endomyocardial biopsy ruled out concurrent myocarditis)	Pneumonitis	β-blocker	Resolution	NR
Ederhy et al., 2017 [22]	NR	Apical ballooning along with mid-ventricular akinesia	+	Sinus tachycardia, T-wave inversion in anteroseptal leads	+ (Trop)	No obstruction	− (cardiac MRI did not reveal any signs of myocarditis)	-	Corticosteroids	Resolution (6 days)	NR
	NR	Basal and median segment akinesia (atypical TTS)	+	T-wave inversion in V2-4	+ (Trop)	Chronic DCA obstruction	Possible myocarditis (diffuse myocardial edema on MRI; biopsy was not performed for exclusion of concurrent myocarditis)	-	Corticosteroids, HF management	Resolution (28 days)	NR
Okamatsu et al., 2020 [14]	Fever, dyspnea, wheezing	Apical ballooning, ventricular hyperconstriction	+	ST elevation in V4-5, T-wave inversion in II-III, aVF, V3-6	+ (Trop)	NR	− (no biopsy or MRI performed for exclusion of myocarditis)	Infusion reaction	Corticosteroids, vassopressors	Resolution (28 days, but died 62 days after due to PD)	-
Anderson & Brooks, 2016 [23]	Chest pain, abdominal cramping, diarrhea (colitis)	Left ventricular dysfunction	-	Widespread T-wave inversion	+ (Trop)	No obstruction	− (no biopsy or MRI performed for exclusion of myocarditis)	-	HF management	Resolution	+
Schwab et al., 2019 [24]	Chest pain, dyspnea	Apical ballooning	+	NR	NR	No obstruction	− (no detailed description of MRI findings for myocarditis exclusion)	Nephritis	Corticosteroids, HF management	Resolution	+
Camastra et al., 2020 [13]	Dyspnea	Apical ballooning (akinesia of mid-apical segments)	+	ST elevation in antero-lateral leads, T-wave inversion V2-4, QT prolongation	+ (Trop)	No obstruction	Possible myocarditis (diffuse myocardial edema on MRI; biopsy was not performed for exclusion of concurrent myocarditis)	Nausea, vomiting	NR	Resolution (8 days)	NR
Norikane et al., 2020 [25]	Dyspnea	Apical ballooning, severe hypokinesis on anterior-septal wall	NR	ST elevation in II, III, aVF, V2-4	+ (BNP)	No obstruction	Myocarditis (cardiac MRI with late gadolinium enhancement and suggestive endomyocardial biopsy)	-	Corticosteroids	Clinical resolution (35 days, but apical ballooning persisted 1 year after)	NR
Singhal et al., 2022 [26]	Diarrhea, confusion, fatigue	Apical ballooning, vigorous systolic contraction of mid zones and anterior and inferior walls	+	ST-elevation	+ (Trop)	No obstruction	− (no biopsy or MRI performed for exclusion of myocarditis)	DKA	Insulin for DKA	Resolution	+
Airo et al., 2022 [27]	Dyspnea, diaphoresis	Apical ballooning, anterior wall and septum akinesis	+	T-wave inversion in V2-4, QT prolongation	+ (Trop & BNP)	Non-significant CAD	− (no biopsy or MRI performed for exclusion of myocarditis)	Hepatitis	Corticosteroids, MI and HF management	Resolution (14 days)	+
‘Sotiria’ case	Chest pain, dyspnea, cardiogenic shock	Decreased contractility of left ventricle with apical ballooning	+	ST-elevation V3-6	+ (Trop & BNP)	Non-significant CAD	− (no biopsy or MRI performed for exclusion of myocarditis)	Hepatitis, skin toxicity	Supportive care for cardiogenic shock, anxiolytics	Resolution (7 days, but 2nd event 6 months after which resolved after 10)	-

BNP: Brain natriuretic peptide; CAD: Coronary artery disease; CAG: Coronary angiography; DCA: Distal circumflex artery; DKA: Diabetic ketoacidosis; HF: Heart failure; ICIs: Immune checkpoint inhibitors; irAE: Immune-related adverse event; LAFB: Left anterior fascicular block; LVEF: Left ventricular ejection fraction; MI: Myocardial infarction; MRI: Magnetic resonance imaging; NR: Not reported; PD: Progressive disease; RBBB: Right bundle branch block; RCA: Right circumflex artery; Trop: Troponin; TTS: Takotsubo syndrome; VT: Ventricular tachycardia; ↑/↓: Elevated/decreased. +/−: Yes/no.

## Data Availability

All data are available in the manuscript, tables, and Appendix A.

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
