# Peer review of "Takotsubo Cardiomyopathy in Cancer Patients Treated with Immune Checkpoint Inhibitors: A Systematic Review and Meta-Summary of Included Cases"

_cancers, 2023, doi:10.3390/cancers15092637_

Round 1
Reviewer 1 Report
Thank you very much for an opportunity to read and evaluated your manuscript.
The paper presents a systematic review of Takotsubo cardiomyopathy in cancer patients treated with immune checkpoint inhibitors. That new data shows that malignancy is an independent risk factor for TTS due to emotional stress. According to review cancer patients treated with immune checkpoint inhibitors with or without chemotherapy experience worse TTS outcomes compered to patients without malignancies suffered from TTS .
It’s a first systematic review for that group of patients with TTS as a cardiotoxic effects of ICIs. Authors analyses 17patients, I suggest collecting more patients for that study. Small control group couldn’t give an ability to draw conclusion for such an important topic.
The discussion describes the significance of different findings such as symptoms, echocardiography findings, ECG changes (including arrhythmias) and type of treatment. Clinicopathologic characteristic of all patients seems to be detailed. Several limitations were also included in study. I find this article very interesting and relevant. The study should be continued in the future. It will be interesting to find results with large scale studies.
Author Response
We thank the reviewer for his/her comments. Our systematic search concluded in a small number of cases (17 patients), which we acknowledge as a limitation in the discussion. Furthermore, we agree with the reviewer that thorough reporting of TTS in the clinical setting and in large scale studies would bring a better insight into the association of the two as critically appraised in the discussion and conclusion of our manuscript.

Reviewer 2 Report
Dear Authors
That is an interesting article with systematic review and meta summary of cases on Takotsubo cardiomyopathy during the treatment of cancer by Immune Check Point Inhibitors.
Authors performed an extended search in published cases and series of cases described in the literature according to PRISMA guidelines up to 13 March 2023.
From seventy-seven articles finally sixteen articles presenting 17 pts were included in the analysis.
This is a remarkably interesting study on presentation of possible myocardial side effects of immune check point inhibitors.
1. Immune checkpoints inhibitors are recognized as therapeutic agents with a strong potential to induce myocarditis –I suggest stressing this point.
2. In your article you described in five out of 17 pts (29%) signs of myocarditis or pericarditis.
In next paragraph 196-198 you wrote, “All patients with available data demonstrated ECG changes suggestive of MI (ST elevation, T-wave inversion [N=16]), had abnormalities of cardiac enzymes (elevation of troponin and/or brain natriuretic peptide 198[N=16]),
Those findings also strongly suggest myocarditis –if this possibility was excluded (biopsy, NMR examination?) or other method –my guess is that in many cases not.
3 Please, if possible, specify, how the possibility of myocarditis was excluded or confirmed (in table 2)- if not analyzed or not ruled out by the Authors pleases state it clearly.
4. Maybe Takotsubo was only a phenotypic presentation of induced by immune check point inhibitors myocarditis –please refer to that point more in your discussion.
Finally, I would like to congratulate all Authors for their interesting work to analyze all those cases published in the literature.
Author Response
We would like to thank the reviewer for his/her invaluable comments.
The reviewer stressed out a very important point, that TTS may be a phenotype/manifestation of immunotherapy-related myocarditis rather a direct, distinct cardiotoxic effect itself.
Indeed, as stressed in the manuscript (lines 336-343), the distinction between the two is difficult and MR studies with gadolinium enhancement or even myocardial biopsy are needed for differential diagnosis and management. Furthermore, we acknowledge that a larger pool of cases with the necessary workup for myocarditis exclusion should be needed to identify whether TTS is a de novo irAE or an epiphenomenon of underlying myocarditis.
As the reviewer suggested, we tried to highlight these remarks in the revised version of our manuscript. Please see the revisions in detail as follow:
- Table 2: We added a comment for the diagnostic workup of myocarditis exclusion for each case (column ‘concurrent cardiac complications’).
- After careful review we identified 6 patients (35%) with concurrent cardiac complications, 5 of which had demonstrated signs of myocarditis (3 with definite diagnosis, 2 with possible diagnosis) (see in table 2 and manuscript lines 343-345).
- From the remaining 11 cases, not reporting concurrent cardiac complications, only two performed (and reported relevant findings) the necessary workup to sufficiently exclude myocarditis (table 2 and lines 345-348).
- All the above have been critically discussed in the revised version of our manuscript (lines 106-108, 131-134, 334-353)

Reviewer 3 Report
The Abstract needs quantification. The Introduction needs more enhancement. Section 4 needs to modified by tree blocks or connectivity chart. Therefore the readers can easily understand the cause and effect of Cancer patients. Well written work
Author Response
We thank the reviewer for his/her comments.
- We have quantified the abstract as advised and according to the Journal’s instructions (lines 29-46).
- We have enhanced the introduction as advised with further remarks regarding the incidence and management of cardiotoxic adverse events (lines 103-108), the diagnosis and management of TTS (lines 114-122), and the difficulties in association of TTS with immunotherapy (lines 123-123, 129-134).
The reviewer commented: “Section 4 needs to modified by tree blocks or connectivity chart. Therefore the readers can easily understand the cause and effect of Cancer patients”.
To our understanding the reviewer would suggest providing a chart summarizing the discussion (section 4) and the main findings of our review. This comprehensive chart would allow the reader to understand the main cause and effect in the development of TTS in immunotherapy treated cancer patients.
Following the useful advice of the reviewer a respective comprehensive chart was drafted to illustrate the rationale and main findings of our study (figure 2 and lines 390-392).
